# Facing the Post-Pandemic Challenges: The Role of Leadership Effectiveness in Shaping the Affective Well-Being of Healthcare Providers Working in a Hybrid Work Mode

**DOI:** 10.3390/ijerph192114388

**Published:** 2022-11-03

**Authors:** Karolina Oleksa-Marewska, Joanna Tokar

**Affiliations:** 1Institute of Management and Quality Sciences, WSB University in Poznan, 61-895 Poznan, Poland; 2Institute of Management and Quality Sciences, Humanitas University in Sosnowiec, 41-200 Sosnowiec, Poland

**Keywords:** healthcare leadership, leadership competencies, healthcare hybrid work, healthcare remote work, affective well-being, job satisfaction, work engagement, work–life balance, meaning of work, COVID-19

## Abstract

The COVID-19 pandemic has brought new challenges to the medical industry, including hybrid work, in which specialists can perform some of their duties remotely, in addition to physical contact with patients and their teams. Hybrid work provides opportunities, but also generates difficulties (e.g., accurate long-distance diagnosis); therefore, there is a need to ensure the well-being of healthcare workers, especially in the context of leadership strategies. As there is little research on leadership practices in remote and hybrid medical worker management, this study analyses the relationship between certain behavioural strategies and competencies of leaders and the affective well-being of hybrid employees. The research was conducted among a group of employees (N = 135) from seven countries who provide healthcare in a hybrid model. The correlations between the variables showed the statistical significance of all leadership strategies introduced into the model and focused on building involvement (employee empowerment and team orientation), creating a shared vision, defining clear goals and strategies, promoting adaptability (change management, promotion of organisational learning and patient focus), managing consistency through shared values, agreement and effective coordination, as well as competencies such as communicativeness, credibility, self-development and digital readiness. Despite the significance of all the relationships, the linear regression showed that the variability of affective well-being was explained mainly by the adaptability leadership strategy. The results of the study expand the knowledge on the competencies of healthcare leaders, and shed new light on the management of medical employees performing remote and hybrid work. Because such research into well-being has not been published to date, the analysis begins an important discussion on redefining leadership in the healthcare sector, taking into account the digital transformation.

## 1. Introduction

The COVID-19 pandemic has brought significant changes in almost all areas of life. In organisations, these challenges include rising stress levels among key employees around the world [1], a greater need to protect the mental health and well-being of employees [2,3] and a transition to remote and hybrid work, which redefines the approach to well-being and mental health. However, the mode of work and the factors that affect the well-being of employees differ depending on the industry. The medical industry has become particularly important because of the pandemic. 

During the pandemic, healthcare professionals suffered from deteriorating health, including symptoms of depression, anxiety and general psychological distress—e.g., burnout [3,4,5,6,7,8,9]. Therefore, protecting the mental health and well-being of medical professionals became a key task of the public health sector [4]. This task should be implemented at the organisational level, primarily by management, as it is mainly the superiors and managers of medical institutions who have an influence on the well-being and health of their employees [7,10,11]. It is especially important to study the leaders’ impact on the well-being of employees in the face of changes such as redefining the model of work from classic to hybrid, which already applies to many medical employees who—in addition to physical contact with patients—can also provide online consultations and perform other duties outside the workplace using telecommunications tools. While remote working gives greater autonomy in time and space, it also requires self-discipline and normative regulations, which leads to putting more effort into work [12]. The transition to the digital work environment has created an urgent need to deal with data overload, employee alienation and increasingly weaker relationships within teams, as well as a declining sense of trust and influence on work [13]. Moreover, hybrid and remote work affect the well-being of medical workers differently than traditional work. During online consultations, it is often difficult or impossible to accurately diagnose a disease (by examining signs and taking accurate measurements), and this may cause more diagnostic errors, patient dissatisfaction and a reduced sense of self-efficacy. This is why it is important to study the leadership strategies that support the affective well-being of healthcare employees who work remotely and in a hybrid model. 

In many countries, the pandemic has stabilised, with protective measures helping to control the number of cases. Although the restriction regimen has been relaxed, we should not ignore the fact that healthcare professionals are facing new challenges and threats—for instance, new waves of COVID-19 and other communicable diseases that may affect society in a similar manner. The hybrid model will most likely remain a standard in the healthcare sector, at least for some professionals. Therefore, we should not only ensure employees’ well-being in relation to the pandemic, but should also redefine leadership and introduce e-leadership elements in order to provide long-term support for the well-being of hybrid medical workers. 

Currently, most strategies improving medical workers’ well-being are aimed at developing individual coping skills [14], while it is necessary to introduce more systemic activities—in management, at the social level and in the organisational culture [10]. These systemic activities include appropriate training and selection of managerial staff based on the competencies and leadership strategies that, in the face of contemporary, post-pandemic challenges, increase the efficiency and improve the health and well-being of employees. 

The latest studies on healthcare leadership during the pandemic primarily relate to dealing with health threats and overload resulting from contact with patients [7,15]. However, there are no studies dealing with leadership in healthcare in the context of remote and hybrid management, changes following digitisation and the challenges brought about by the pandemic [16,17,18,19], which require a number of skills, including knowledge of and openness towards health information technologies. This frequently tends to fail, largely due to poor leadership in healthcare organisations [20]. Therefore, there is a need to expand research in remote healthcare leadership [16,19], especially to investigate which healthcare leadership strategies and styles can improve the well-being of employees [17,21,22], and which leadership strategies are important in case of digital transformation [23]. This paper also contributes to the comparison of respondents from different countries, as other researchers noted that samples from American or European regions where healthcare workers are likely to face psychological distress, especially in light of pandemic challenges, need further investigation [17].

As there is scarcely any research into the issue of well-being and possible organisational strategies for improving the quality of hybrid and remote work in the medical industry, we designed a study to deal with these issues. Thus, the article addresses an important aspect of leaders’ behavioural strategies as resources for workers to cope with the demands of hybrid work and to remain well. The results fill the cognitive gap in this field, and the study begins an important discussion on the search for coherent, effective leadership strategies that are appropriate for new challenges in the healthcare system. We decided to answer the following research questions: RQ1) What leadership practices and competencies are relevant for healthcare management, especially in light of remote and hybrid work? RQ2) Which leadership practices and competencies influence the affective well-being of remote and hybrid healthcare specialists? To answer these research questions, we analysed different leadership models applicable to the healthcare system. We chose the Denison leadership effectiveness model as it is focused on four different clusters of strategies, involvement, consistency, adaptability and mission [24], aimed to strengthen the flexibility of the company, which is crucial in hybrid work. We also analysed four competencies, communicativeness, credibility, self-development and digital readiness, which we added to the model as important capabilities enabling effective leadership practices in remote work [25]. In the methodological step, Spearman’s correlation coefficient between different leadership strategies and the level of affective well-being was analysed, and in order to determine which leadership strategies explain the variability of affective well-being, stepwise multiple regression was used. According to the research findings, although all correlations between the chosen variables were significant, well-being is mainly explained by the adaptability cluster, which is connected with patient orientation, a proactive approach to changes and promoting organisational learning. As a contribution to existing knowledge, we propose an empirically verified model of healthcare leadership strategies essential to hybrid employees’ well-being. To the authors’ knowledge, such models are theoretical or devoted to the stationary work of healthcare providers. The findings and proposed model bring practical implications for healthcare management development, indicating which competencies and managerial behaviours are worth improving in the face of post-pandemic challenges in maintaining the well-being and high productivity of employees. 

To present the research findings and their relevance, the paper covers following sections: theoretical framework based on the job demands–resources (JD-R) model, a literature review analysing the well-being of healthcare providers from the perspective of affective components (relevant while performing emotional labour) and finally the leadership strategies and models applicable to the healthcare context and connected with the well-being of employees. The theoretical part of the article is concluded with hypotheses, which are then verified in the empirical section, presenting the research design, sample, research procedure, the performed statistical analyses and the obtained results. In the discussion, we point out the importance of the chosen leadership strategies and competences, and explain why adaptability turns out to be the most important cluster of leadership strategies in the context of increasing the well-being of remote and hybrid healthcare specialists. The discussion culminates in the proposal of a leadership strategies and competencies model, which can be practically used in training healthcare managers, as well as improving the management practices aimed at enhancing hybrid employees’ well-being.

## 2. Theoretical Framework and Literature Review

The authors have chosen the job demands–resources (JD-R) model as a theoretical framework. It postulates a relationship between job demands (the level of job content risks) and job resources (personal and organisational factors allowing workers to cope with job demands), with consequences for the health and well-being of employees, as well as for management practices [26]. Leaders’ behavioural strategies can be regarded as organisational resources [27] that influence the ways in which employees deal with job challenges, thus achieving a higher level of well-being. Other conceptualisations present leadership strategies as an independent factor that affects both resources and demands [28], and therefore leadership effectiveness is a valuable variable to explain the well-being of healthcare workers. The model has already been used by other researchers to assess the relationship between leadership in healthcare and the level of psychological well-being [29]; it therefore seems to be an appropriate theoretical framework.

## 3. Well-Being of Healthcare Providers

Well-being is a multidimensional construct that, depending on one’s perspective, can be defined by different variables. In this study, the domain-specific affective well-being perspective [30] was adopted, which means that the focus of the study was on the work-related emotional well-being of medical workers. Healthcare professionals, who have contact with patients/clients, usually perform emotional labour, i.e., work that requires demonstration of the emotions determined by organisational procedures, while masking or modifying one’s real feelings in order to avoid a negative impact on the patient/client. According to Arlie Hochschild’s theory [31], emotional labour is characterised by direct contact with a patient/client (face-to-face or voice-to-voice) and evokes the desired affective states (e.g., gratitude, relief or calmness). Through management and training methods, a superior may influence the emotional labour performed by their subordinates (e.g., by rewarding cordiality towards clients/patients). Therefore, when considering the well-being of healthcare professionals—especially in the context of hybrid and remote work—its affective dimension related to emotional states is important. As part of the affective well-being analysis, particular attention was paid to the dimensions associated with organisational factors, including job satisfaction—which is a reflection of the positive emotions resulting from job experiences—with affect and other job attitudes being key components [32]. Satisfaction is one of the most often discussed subjects connected with employees’ organisational behaviour. It is usually perceived as the attitude that results from the individual’s reaction to the surrounding work conditions [33]. In the affective model of well-being, Horn et al. [34] assumed that job satisfaction is related to affective well-being, as it consists of affective, cognitive and behavioural components and largely reflects the assessment of emotions on the pleasure–displeasure scale, which is important for well-being [35]. Job satisfaction in healthcare is strongly associated with the level of overall well-being and the evaluation of organisational conditions [36], Thus, it is an important component of the assessment of employees’ emotional states.

Additionally, work engagement was taken into account, perceived as a work-related positive attitude characterised by vigour, absorption and dedication to work [37], which is a persistent affective/motivational state [38]. Ensuring the engagement of healthcare professionals is extremely important for improving their well-being and for systemic activities [10]. The meaning of work, i.e., the perception of work as significant and relevant, was also taken into consideration, with the related content and qualities, organisational goals, culture and relations with colleagues and the community [39]. Research into medical staff has shown that the meaning of work has a significant impact on the levels of retention [40] and emotional exhaustion [41,42], and this even further indicates the importance of including this variable in the assessment of healthcare professionals’ well-being when performing emotional labour. 

The last element of affective well-being is the evaluation of one’s work–life balance, which is significantly influenced by remote work [43]. Until now, when healthcare workers usually performed their duties outside the home, their work–life balance could be disturbed by long periods of on-call duty and difficulties resting after contact with patients. However, in remote and hybrid work, some duties are brought home or elsewhere, outside the office or institution; working time becomes more flexible, which creates new difficulties maintaining a balance between work and private life.

## 4. Leadership Effectiveness and Its Influence on Well-Being in Healthcare

In the turbulent times of the pandemic, the strong leadership of managers is important in order to achieve results, despite external difficulties. Organisational goals can be effectively met by caring for the most important capital of the organisation: its employees. They are responsible for the development of the organisation [44], so their well-being is crucial. Therefore, we have undertaken an analysis of leadership from the perspective of effectiveness, understood as a process in which managers interact with employees so as to achieve organisational goals and other outcomes—for instance, managing changes, building commitment to group objectives and increasing psychological well-being and job satisfaction [45]. Effective leadership has a strong impact on employees’ well-being, engagement and emotional states [10,29,46,47], and consequently on the performance and effectiveness of the whole institution.

Healthcare leadership is the ability to effectively and ethically influence others for the benefit and sake of patients [48]. Effective leadership in healthcare is critical for improving the health and well-being of workers, and thus for implementing healthcare reforms [49]. At present, healthcare leaders must be flexible, be able to clearly communicate their vision and goals and be compassionate and open to dialogue [50] and have knowledge and the willingness to, for instance, implement ICT solutions, thus creating a long-term vision that increases the sense of stability among employees, ultimately leading to the organisation’s success [51]. Moreover, healthcare leaders are expected to be agents of change and innovators [49]. The reality forces leaders into continual development and, above all, self-improvement, understood as a process in which everyone individually takes the initiative, with or without others’ help. Such self-improvement requires that the leader chooses a defined direction of action and has knowledge of management strategies that will contribute to success. Despite the growing awareness of the key role of leadership in improving the quality of healthcare, the knowledge of specific competencies and leadership strategies leading to success is limited and requires exploration [48]. 

The literature provides lists and models of effective leadership strategies adopted by superiors in the health service sector. However, these strategies are usually presented in the context of overall performance and effectiveness, and less often that of the well-being of employees. K. Groves and A. Feyerherm reviewed research and best practices for assessing overall leadership potential in healthcare; based on the theory and research on leadership potential, they identified four leading components and fourteen criteria for leadership potential in healthcare organisations. The model consists of the following primary components: (1) analytical aptitude, (2) learning agility, (3) leadership capability and (4) people savvy. The horizontal dimension includes (a) cognitive competencies that require thinking, problem-solving and learning from dynamic environments and (b) behavioural competencies, based on skills that condition the ability to influence, inspire and engage others. The vertical dimension of the model consists of leadership potential factors that affect (a) the macro-level—for example, a healthcare institution or system and the wider industry—and (b) the micro-level of self-management skills, interpersonal skills and the capacity to influence others [52]. Another noteworthy comprehensive model of leadership strategies, the Duke Healthcare Leadership Model, was proposed by Hargett et al. [48]. The model is based on competencies recognised as the most important attributes of effective leadership in healthcare, the most significant of which is patient centredness, which distinguishes healthcare leadership from leadership in other areas. The model covers patient-centredness and the core competencies of (1) emotional intelligence, (2) integrity, (3) selfless service, (4) critical thinking and (5) teamwork.

Among specific behavioural strategies that increase the effectiveness of leadership in healthcare, the following areas were indicated: building trust, promoting employee commitment to joint action and promoting values [53]. In a research report presented by Kane et al. [54], it was concluded that the most effective healthcare leaders have the following skills: direction (providing vision and purpose), innovation (conditions for experimenting), execution (empowering employees), collaboration (across boundaries), inspirational leadership (making people follow), business judgment (making decisions in uncertainty), building talent (self-development) and influence (persuading and influencing stakeholders). A similar but more in-depth catalogue was proposed by Kumar and Khiljee [55], pointing to the importance of credibility and confidence in the leader when the vision and long-term goals are communicated. The authors also indicated the ability of the leader to engage the team in developing a common approach to work and in building lasting commitments; they mention supporting creative participation, having a common, inspiring goal, developing the capacity of individuals and entire teams in the long term, focussing on innovative changes, being ready for continuous improvement and taking up bold challenges. In the face of the COVID-19 pandemic, the following strategies have been identified as being particularly important: the ability to stay calm, good communication, cooperation, coordination and providing psychosocial support to the staff [56].

There are a few, less comprehensive studies on managers’ competencies and behavioural strategies that improve the well-being of medical workers. Many studies have provided general conclusions—for instance, an improvement in healthcare workers’ well-being was observed under participatory management [57], good team management and communication [58], as well as relational leadership and the ability to empower employees [21]. Some models of leadership strategies related to well-being are theoretical, such as the Model of Leadership Influence on Health Professional Well-Being, which includes leadership strategies such as ensuring basic human needs, self-care of employees, active listening, addressing concerns, advocating, connecting employees with resources, scheduling adjustments and conducting regular well-being debriefs [59]. Other leadership strategies that have had an impact on well-being in healthcare during the pandemic include the ability to anticipate and deal with crises, knowing how to reduce stress and helping to maintain a work–life balance among employees [2]. Better well-being in this professional group may also be observed after the introduction of e-leadership elements into management, i.e., skilful leadership via electronic channels [60].

Even though studies indicate the importance of leadership for the well-being of healthcare workers, the conclusions are often general and rarely point to specific managerial strategies. Therefore, we attempt to select a set of strategies to increase leadership effectiveness in enhancing employees’ well-being, especially in hybrid and remote work.

In order to verify which leaders’ behavioural strategies increase the level of affective well-being, the Denison Model was adopted, which focusses directly on leadership strategies that influence organisational effectiveness and involve the main organisational culture traits: involvement, consistency, adaptability and mission [24], However, we have noticed the need to supplement the model with leadership competencies that are fundamental and universal for all areas and that enable the implementation of behavioural strategies within the four leadership traits. Therefore, we add four additional competencies [25], which, in our opinion, hold together all dimensions of leadership effectiveness, especially in the context of remote work. As today’s leaders of hybrid teams need to focus on building trust and credibility, taking care of the dynamics of organisational development, virtual team cooperation and effective communication, the universal competencies include communication—one of the basic and most important managerial competencies in healthcare [22,50,51], which, according to healthcare workers, improves their well-being [3,61]—and credibility, also considered to be the foundation of effective leadership in healthcare because it allows for ‘getting things done’ by building recognition and integrity [22] and because it improves the affective well-being of medical workers [3]. Self-development was also added to the model as a basis for novel development opportunities and openness to new solutions. It is also a key competence that increases the effectiveness of leaders in healthcare [54]; along with digital readiness, understood as cognitive openness to new technologies, a proactive approach to digital transformation and a focus on learning new methods of communication and cooperation in the virtual world [25], it is a response to the need for new technologies in healthcare, which should be openly approached by leaders [51]. The conceptualisation of leadership effectiveness strategies, including the four additional competencies, is presented in Table 1.

A literature review revealed that leadership effectiveness is of great importance in shaping the affective well-being of healthcare workers employed in a hybrid model. Therefore, it is crucial to recognise which behavioural strategies increase the effectiveness of leaders in promoting well-being. Given the above, the following hypotheses were formulated:

**H1.** 
*The level of well-being increases along with leadership effectiveness.*


**H1a.** 
*The higher the effectiveness in terms of the adaptability strategies, the higher the affective well-being.*


**H1b.** 
*The higher the effectiveness in terms of the involvement strategies, the higher the affective well-being.*


**H1c.** 
*The higher the effectiveness in terms of the consistency strategies, the higher the affective well-being.*


**H1d.** 
*The higher the effectiveness in terms of the mission strategies, the higher the affective well-being.*


**H1e.** 
*The higher the effectiveness in terms of the core remote competencies, the higher the affective well-being.*


Given the fact that most summaries of key leaders’ competencies and strategies in the face of today’s challenges mainly relate to increasing the commitment and empowerment of healthcare workers (relating to the involvement dimension), as well as the ability to act flexibly, create a clear vision, focus on the patient and adapt to changing conditions and expectations (corresponding to the adaptability dimension), the following hypothesis was formulated:

**H2.** 
*The variability in affective well-being is most explained by the strategies of adaptability and involvement.*


The conceptualization of variables is presented in Figure 1.

## 5. Research Methodology

To verify the stated hypotheses, primary data were collected through an accurately planned research design. It was crucial to find respondents meeting two criteria: working in a hybrid mode and being a healthcare specialist. It was also important to have an international sample so as to be able to draw conclusions in an extended context. Because of this, non-probability sampling was chosen, and the selection of respondents was done with help from reliable research agencies operating in six different countries. Respondents were asked to complete a survey composed of five questionnaires (one connected with the assessment of leaders’ behaviours and competencies and four validated scales devoted to affective well-being components). Subsequently, quantitative data emerging from conducted surveys were analysed through statistical methods. As a consequence, results obtained were interpreted, and a leadership model showing strategies and competencies enabling an increase in the affective well-being of healthcare providers was proposed. The methodological process is illustrated in Figure 2.

## 6. Methods and Sample

### 6.1. Sample

The study involved 135 medical professionals providing hybrid healthcare within various specialisations. The respondents were requested to indicate the area of their work and their job title. Some of them entered a general name, such as ‘doctor’, ‘specialist’ or ‘healthcare’, while others specified positions that included paediatricians, internists, laboratory technicians, nutritionists, pharmacists, psychologists, clinical physicians, psychotherapists, endocrinologists, physiotherapists, midwives and one specialist each in emergency medicine, nursing, optometry, phlebology and dentistry. The respondents came from seven European countries (United Kingdom, Netherlands, Poland, France and Spain) and North America (United States and Canada). Women were more numerous (n = 103) than men (n = 32); the average age of the respondents was 39.6 years. The work experience of the respondents ranged from less than one year (10 respondents) to 50 years (1 respondent), while the average length of work experience was 8.8 years.

### 6.2. Procedure and Ethics

The research was conducted from February to March 2022. The respondents, who completed the survey electronically, were selected for the sample using the purposeful selection method (criterion: hybrid/remote work in the field of medicine), with the help of external research agencies. The respondents completed the survey in their native languages (all scales were translated and reverse-translated by native speakers). The study was designed based on the ethical guidelines of the WMA Declaration of Helsinki (1964 with later amendments), according to which the respondents were informed about the right to withdraw from the study at any time, without giving any reason. Before starting the study, the respondents were asked about the mode of their work (the research script automatically rejected people who selected the ‘traditional’ mode). They were then requested to provide informed consent to participate in the study, which was explained and described in the manual. The subject of the research was outlined, as well as the quantity of questionnaires and the possible psychological consequences of completing the survey (emotional states related to the assessment of individual aspects). The study design was approved by the academic ethics committee of the University of Lower Silesia.

## 7. Measures

### 7.1. Leadership Effectiveness Strategies

In order to assess the managers’ behavioural strategies, a Leadership Effectiveness Index was applied, which was the shortened version of the Denison Leadership Development Survey, for which validation studies showed a Cronbach’s alpha value greater than 0.90 for each of the four traits [24]. Although the Leadership Effectiveness Index (LEI) has not yet been validated, after consulting with the authors of the questionnaire, we decided to use the LEI version because it consists of 12 items selected through statistical analysis among the 96 items of the original questionnaire. Since the LEI is shorter than the original, the time to complete the study was not prolonged, which allowed for greater reliability in completing the survey. The LEI verifies four groups of behavioural strategies: involvement (items related to empowerment, team orientation and organisational capability), with a Cronbach’s alpha value in this study of 0.85; consistency (following core values, promoting engagement and coordination), with a Cronbach’s alpha of 0.82; adaptability (creating changes, promoting organisational learning and being customer/patient-focussed), with a Cronbach’s alpha of 0.8; and mission (creating shared vision and defining goals and main strategies), with a Cronbach’s alpha of 0.83. We added four questions in order to verify the selected universal, core competencies of remote/hybrid workers’ leaders: communication, credibility, self-development and digital readiness. The Cronbach’s alpha value for core competencies was 0.83.

### 7.2. Affective Well-Being

Affective well-being is a latent variable, which covers the assessment of job satisfaction, work engagement, meaning of work and work–life balance. Analysing the adequate research design of work-related well-being measurement [63], we used four different, reliable questionnaires. Job satisfaction was measured on the five-item scale called the Brief Job Satisfaction Measure, proposed by T. Judge et Al., which showed satisfactory reliability (0.88) [64]. In the current study, the Cronbach’s alpha value was 0.79. Respondents gave their answers on a 7-point Likert scale; two statements were reversely scored (“Each day of work seems like it will never end” and “I consider my job rather unpleasant”), and the higher the total score, the higher the job satisfaction indicated by the respondent. Work engagement was measured using the nine-item questionnaire UWES-9, developed by W. Schaufeli and A. Bakker [37] (Cronbach’s alpha for the total scale in validation study varied between 0.85 and 0.92 [65]). The UWES-9 enabled us to test three components: vigour (Cronbach’s alpha in current study: 0.85), absorption (Cronbach’s alpha: 0.66) and dedication (Cronbach’s alpha: 0.84). The UWES-9 has a 6-point scoring scale, no reverse items and enables us to assess the general level of engagement as a mean score of all three subscales (components). Although authors present norms for nine national samples, in the current study, the level of engagement was not categorised as it was not relevant to the hypothesis testing. The meaning of work was measured using a subscale from the third version of the Copenhagen Psychosocial Questionnaire, called Meaning of Work (reliability due to different studies varied between 0.68 and 0.74 [66]), which consists of two questions (“Is your work meaningful?” and “Do you feel that the work you do is important?”). Respondents answered these questions on a 5-point scale, from “to a very small extent” to “to a very large extent”. The Cronbach’s alpha for this scale in our study was 0.8. Work–life balance was tested with a single item (“the demands of my work interfere with my home and family life”) from the General Nordic Questionnaire for Psychological and Social Factors at Work (QPS Nordic) [67]. Respondents could assess this item on a 5-point scale, from “never/barely never” to “always/almost always”. In total, respondents had to assess 17 items in four different questionnaires. The affective well-being variable was computed through the CFA technique. Before final calculations, the QPS Nordic item was inverted because, unlike other scales, the higher the score, the worse the level of work–life balance. Subsequently, for each observation in the dataset, a predicted value for the affective well-being variable was computed based on factor analysis.

The Cronbach’s alpha value for the variable of affective well-being in this study was 0.8.

## 8. Statistics

Statistical analysis was conducted with the use of the software programme R; regression was analysed using the ‘base’ package and diagnostics was performed with the ‘car’ package. The following analyses were conducted:

Descriptive statistics, including frequency, percentage, mean and standard deviation of the variables; the correlation between leadership effectiveness traits and affective well-being, using Spearman’s correlation coefficient and stepwise multiple regression, to study the leadership traits that explain the level of affective well-being.

### 8.1. Descriptive Statistics

Leadership effectiveness strategies and core remote competencies were adopted as independent variables in the analysis; each strategy was considered separately (12 strategies and 4 key competencies). The variables were analysed as the following main traits: leadership consistency, leadership involvement, leadership adaptability, leadership mission and core remote competencies. Affective well-being was adopted as a dependent variable.

The normality of the variables’ distribution, assessed using the Kolmogorov–Smirnov test with Lilliefors correction, showed that the variables did not have a normal distribution. The mean values of the independent variables oscillated around 4.5, while the mean for affective well-being was 4.31. Descriptive statistics are shown in Table 2.

### 8.2. Correlations between Independent Variables and Affective Well-Being

In order to verify the relationships between the variables, the strength and significance of the correlation between managers’ strategies and affective well-being were measured; then, the correlations were determined between the level of affective well-being and the four types of leadership, based on the Denison model and including the core remote competencies added to the model. Because the variables did not have a normal distribution, the correlation analysis was based on Spearman’s rank correlation. The correlations between all independent variables and affective well-being are presented in Table 3.

The analysis showed that all 12 leadership effectiveness strategies and four core remote competencies were significantly correlated with the level of affective well-being of the healthcare professionals. Because all relationships were directly proportional, with a better assessment of leaders’ behaviour and competencies, the employees’ affective well-being grew. The strongest correlations were shown for the ability to coordinate and focus on patients, while the weakest correlation was observed for self-development. Based on the results, hypothesis H1 was accepted. 

Next, the correlations were determined, being divided into leadership effectiveness strategies for the following traits: involvement, consistency, adaptability, mission and core remote competencies. An increase in the values for all traits was associated with an increase in the values for the variable of affective well-being. The results of the analysis are presented in Table 4.

All correlations between the level of affective well-being and certain leadership traits were statistically significant and of moderate strength; therefore, the results allow for hypotheses H1a through H1e to be accepted.

### 8.3. Regression Analysis

In order to verify which of the leadership strategies explain the variability in the affective well-being of the respondents and to what extent, stepwise multiple regression was conducted by introducing traits of leadership effectiveness and core remote competencies as independent variables. Initially, an analysis of multicollinearity was carried out. The variance inflation factor (VIF) for each variable was between 3.42 and 5.72, and values less than 10 indicate that involvement, consistency, mission, adaptability and core remote competencies are not collinear.

The Breusch–Pagan test was applied to verify the assumed homoscedasticity of the variance. The results showed that the variance of the residuals was different, indicating that the homoscedasticity of the variance (chi^2^ = 8.77; *p* < 0.01) was not confirmed. Although residuals in the model decreased or increased along with an increase in the predicted variable results, these differences were negligible. A Kolmogorov–Smirnov test with Lilliefors correction was conducted to verify the normality of the residual values’ distribution for the model as tested. The analysis showed that the distribution of the results for regression model residuals was statistically similar to the theoretical normal distribution (KS = 0.07; *p* > 0.05). The results regarding the normality of distribution and multicollinearity are presented in Table 5.

Next, the regression analysis showed the statistical significance of the model (F = 3.51; *p* <0.01). The unadjusted and adjusted coefficients for the explained variance were 0.12 and 0.09, respectively. The analysis of R² revealed that the regression model of independent variables explained approximately 12% (9% after adjustment) of the variability in affective well-being, which is low. The analysis of the statistics for individual predictors in the model showed that the variability of affective well-being explained only one trait of leadership: adaptability (β = 0.42, *p* < 0.05). The regression results are presented in Table 6.

The regression analysis partially confirmed hypothesis H2: the trait of adaptability alone explained the variability in affective well-being, while involvement did not increase the variance in well-being.

## 9. Discussion

Using the job resources–demands model, the authors aimed to expand the literature on healthcare leadership by identifying the relationship between certain leadership strategies and competencies perceived as organisational resources for balancing the demands of hybrid work that have an impact on the affective well-being of healthcare employees. In the face of the COVID-19 pandemic, the medical industry has not only experienced overload, stress and health risks, but also changes in the mode of work: many employees now perform some of their duties using ICT tools. In the face of these changes, which will most likely become the standard for many healthcare professionals, it is important to indicate specific management strategies to increase leadership effectiveness and promote better well-being among employees.

The correlation between the selected leadership strategies and the level of affective well-being showed that all the strategies and competencies had a significant influence on well-being. This demonstrates that the adopted research model is a very good fit to analyse employees’ well-being. The strategies studied using the LEI (the shorter version of the validated Denison Leadership Development 360 survey [24]) were applied as a basic model that was derived from the theory of the Denison Organisational Culture Model, which indicates the importance of the connection between leadership effectiveness and organisational culture. In the model, 12 practices that are crucial for effective leaders were selected and grouped into four main traits: involvement (empowerment, team orientation and organisational capability), consistency (promoting core values, increasing engagement and coordinating different processes), adaptability (creating changes, promoting organisational learning and being customer/patient-focussed) and mission (creating a shared vision and defining goals and main strategies). The study was extended by four core competencies related to remote management: communicativeness, credibility, self-development and digital readiness [25].

The results confirm those of other studies demonstrating the importance of the following leadership practices for healthcare workers and their well-being: employee empowerment [21,22,68,69], team orientation [10,70,71,72], organisational learning [52,72], effective coordination and defining goals [52,56,72], creating a clear vision [49,51,53,71,72,73,74], increasing engagement [10,53], creating and responding to changes [10,52,53,74] and patient focus [48,72].

There was also a significant correlation between the four core remote competencies. These competencies constitute the core of contemporary leadership, indicating readiness for the challenges of remote and hybrid work, as they enable the practices specified in the Denison model to be implemented [25]. The importance of these competencies for improving the well-being of healthcare professionals was also reported by other studies. Communicativeness is a very important skill, even a crucial one, in the context of healthcare leaders’ tasks [52,56,71,72,73], especially in terms of communicating vision in remote work [51]. The second competence, credibility—both personal and professional—also plays a significant role in building trust and commitment among healthcare workers [10,22,52,70]. This is similar to self-development, understood as being open to the search for new solutions and acquiring industry knowledge [52], which can be creatively transformed by leaders to improve the functioning of the healthcare sector [70]. The last competence is digital readiness, which is specific to remote and hybrid management. This competence refers to an openness to technological solutions and a readiness to implement and use them in order to improve the work of both one’s superiors and subordinates [25]. Other studies have also indicated the importance of this competence for contemporary healthcare leaders [20,51,74]. The relationship between the digital readiness and affective well-being of hybrid employees was moderate; this may have resulted from the fact that the respondents primarily contacted patients rather than their superiors through remote working tools; therefore, a manager’s readiness to implement technological solutions does not have such a great impact on their emotional state as the use of IT tools in a different context.

In addition to correlation analysis, linear regression was performed in order to test which of the leadership strategies and competencies explained the variability in affective well-being. The results indicated one important trait—adaptability—and the practices that develop this trait. Adaptability involves translating the demands of the environment into actions; adaptable leaders have abilities and beliefs that help them to receive and interpret signals from the environment and respond to them properly, so that the organisation can introduce internal changes that enable growth and development [24]. The results show that in these turbulent times, healthcare workers’ well-being increases when they feel that their leaders are ready to face challenges and are able to direct organisational activities in line with what is required, including the demands of patients. This was confirmed by the results showing that combining adaptive capabilities and agility with logical actions and good coordination is crucial in the long-term fight against the effects of the pandemic [50,56]. The variable of adaptability contained the following strategies: creating changes and constantly searching for better ways to work, dealing constructively with failures and mistakes and focussing on patients. These strategies may be the most important for promoting affective well-being because they respond to the needs resulting from the changes caused by the pandemic. The strategies provide the opportunity to reduce emotional tension (especially a constructive approach from managers towards employees’ mistakes) and help workers to cope with the new reality of dealing with patients, the requirements of which have also changed and are more difficult without direct contact (e.g., making an accurate diagnosis or choosing appropriate medications via online consultation alone). The results also confirm the key role of focussing on the patients’ welfare as a characteristic feature of leadership models for healthcare [48]. According to Ref. [37], care for the patient, their safety and their overall experience is the main determinant of the quality of healthcare. Therefore, effective leadership is crucial for improving the quality of healthcare and is necessary to face new the challenges of patient care [70]. Adaptability is also about creating changes and promoting organisational learning. Readiness to proactively respond to changes in the environment and healthcare requirements and a focus on continuous education, both at the individual and organisational level, are competencies of healthcare leaders that have also been noted in other studies as important variables related to the efficiency of the entire system [52,70].

Contrary to theoretical assumptions and the significant correlations shown in the research, the variable of involvement did not significantly explain the variability of affective well-being. The result is quite surprising in the face of research into management strategies that are crucial for the quality of healthcare and employees’ well-being, where the role of workers’ involvement, team building and promoting cooperation are often emphasised and translated into better well-being, engagement and job satisfaction [10,52,53,70]. However, the results may indicate differences in the mode of work. When performing hybrid work, less time is spent with the team and in direct contact with one’s superior; thus, in terms of affective well-being, competencies related to building and strengthening the team are less important. It is more important to skilfully adapt to the requirements of the environment, adopt a proactive approach to changes and organisational learning and adopt a patient-focussed attitude (specialists providing hybrid healthcare meet patients in a different, less comfortable situation). This can be explained with the theoretical model of job demands–resources. In the face of increasing demands (the new reality of dealing with patients, less stable working conditions, etc.), employees firstly focus on the resources that help them to improve their emotional well-being.

It is worth mentioning that the regression model somewhat explains the variance in affective well-being (12%), which indicates a number of factors not covered by the model that affect the level of well-being in hybrid work (factors other than leaders’ behaviour and competencies). However, in the context of exploring the still relatively unknown and very important area of remote and hybrid management in healthcare, the analysis of these results is valuable and provides significant practical conclusions. Therefore, the authors used the results to achieve the conceptualisation of an explorable model indicating a set of leaders’ competencies and behavioural strategies that are important for the well-being of workers providing hybrid healthcare (see Figure 3).

Our findings bring implications for both the theoretical and practical context. Our results contribute to the existing literature by proposing the conceptualisation of hybrid healthcare employees’ affective well-being, and systematising leadership models important for the healthcare sector. We fill the indicated cognitive gap by proposing the leadership effectiveness model, enhancing the well-being of hybrid healthcare providers. As to the practical implications, our findings and the proposed leadership strategies model can be used in the training process of healthcare managers, aiming to improve leadership skills relevant in enhancing employees’ well-being and productivity under the current, turbulent post-pandemic circumstances. As healthcare systems require more flexibility to adopt a digitisation perspective [16,19,23,49,51], and a more patient-oriented focus [48,72], the proposed leadership model could be an important step in shaping more open and adjustable attitudes among healthcare managers.

## 10. Limitations and Further Research

It is recommended to take into account some limitations that result from the research methodology. The first limitation is related to the sample. It would be worth enlarging the study group and categorising the respondents according to their profession or the number and type of professional tasks that they perform remotely. The level of well-being and the assessment of superiors’ competencies may differ depending on whether the respondents work directly with patients and perform administrative or educational work remotely, or whether they deal with patients both directly and using telecommunication tools. In addition, in order to more closely study the assessment of the superiors’ competencies and behaviour, it would be worth expanding the research with additional, validated tools. Although the shortened version of a validated questionnaire was applied, the internal validity was high (Cronbach’s alpha value above 0.8); therefore, the tool was considered sufficient to measure the superiors’ subjective assessment.

Another limitation of the study results from the cross-sectional nature of the measurement, which can complicate the analysis of cause-and-effect relationships between the variables. Therefore, it may be troublesome to explicitly indicate whether leaders’ strategies have an impact on affective well-being or whether people with a higher level of well-being assess their superiors better and appreciate their competencies. Nevertheless, as stated before, leadership effectiveness strategies have been proven to be significant predictors of employees’ well-being, and, therefore, it seems justified to choose a conceptual model of variables in which leadership effectiveness strategies and competencies are adopted as independent variables.

Another limitation of the study is the use of self-reporting research questionnaires. This method of measurement carries the risk of overestimating correlations between the constructs, because of the common-method variance [75]. However, in the case of identifying attitudes and subjective emotional states, it is difficult to find a more adequate method of measurement.

Another limitation of the study is the results of the regression model. Although affective well-being was only somewhat explained by the variability in the assessment of leadership strategies (R^2^ = 0.12), it was included in further analysis as a clue to investigate the challenges encountered in the face of the pandemic and in hybrid work. Given the above estimates, we believe that the effect is not trivial, but weak; therefore, further investigation is needed to establish stronger relationships and more precise predictions. It would be also beneficial to conduct research to verify the proposed model through a structural equation modelling (SEM) analysis.

## 11. Conclusions

The conducted research, along with the literature review, helped to identify important leadership practices and competencies in shaping the well-being of healthcare professionals working remotely and in hybrid mode. It is worth mentioning that hybrid work requires different strategies to stationary work in terms of building satisfaction, engagement, meaning of work and employees’ work–life balance: our research shows that, in particular, being adaptable as a leader and helping employees to react correctly to changes and patients’ needs is great support in building their well-being.

This study offers an introduction to the important discussion on modern leadership in healthcare. It can also be used to search for a model of leadership competencies for higher effectiveness and well-being among healthcare employees that would be comprehensive and verified, both theoretically and practically. This would fill the cognitive gap in the research on remote leadership in healthcare and prepare medical leaders for current and future challenges in the face of the pandemic.

## Figures and Tables

**Figure 1 ijerph-19-14388-f001:**
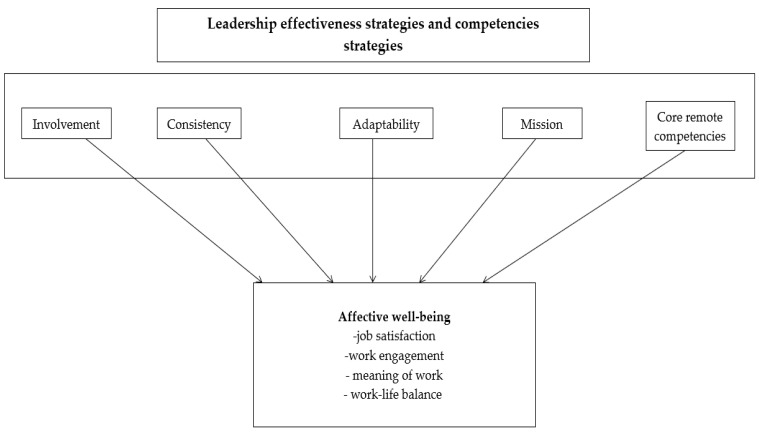
Variables used in the study.

**Figure 2 ijerph-19-14388-f002:**
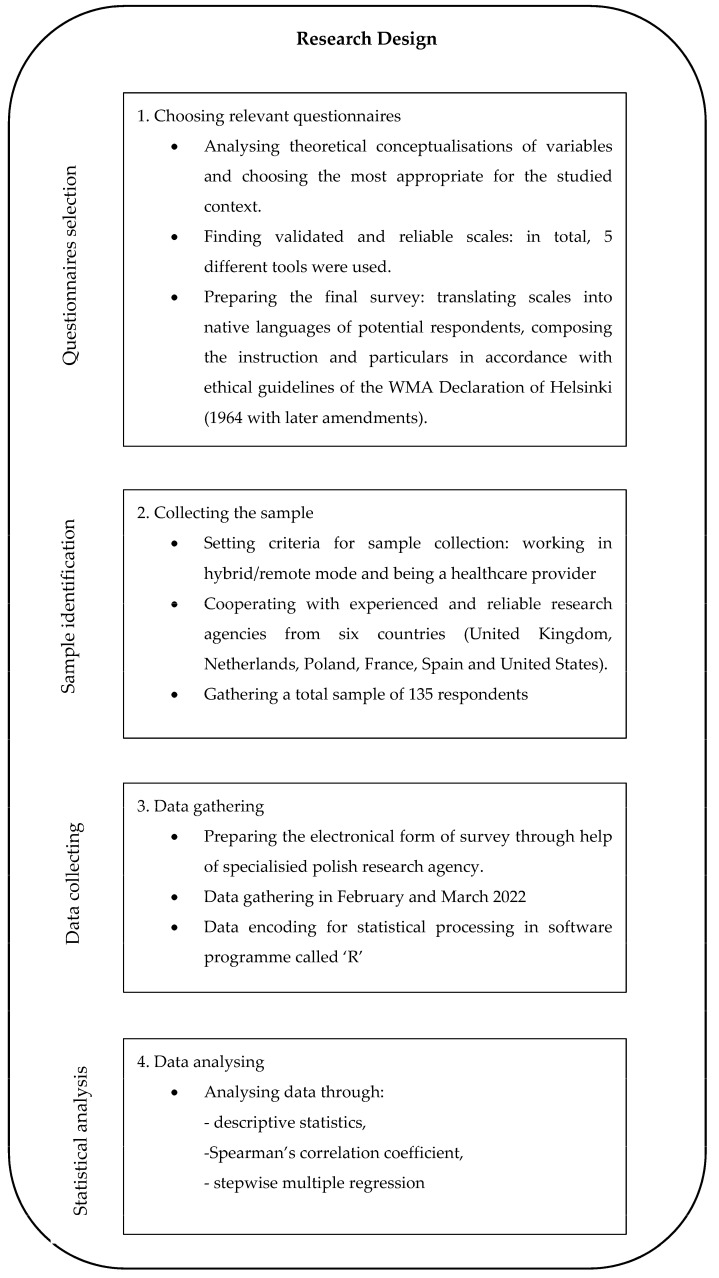
Research design of the current study.

**Figure 3 ijerph-19-14388-f003:**
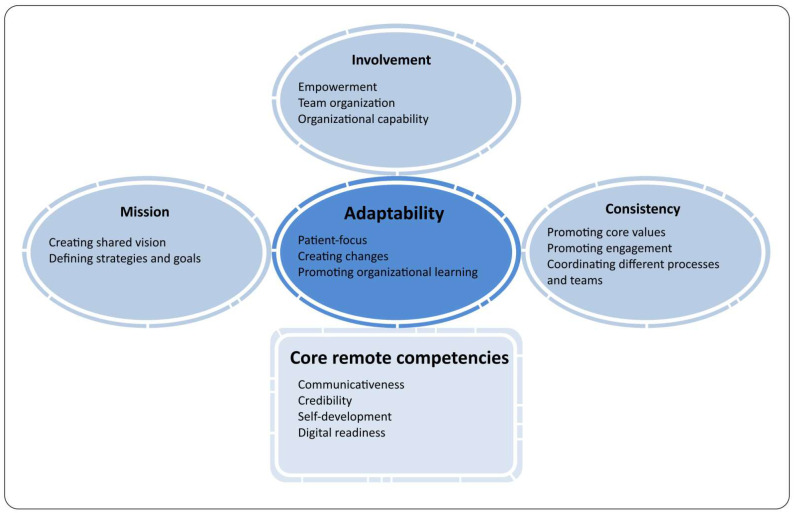
The model of healthcare leadership strategies important to hybrid employees’ well-being.

**Table 1 ijerph-19-14388-t001:** Leadership effectiveness strategies and core remote competencies.

Leadership Effectiveness Strategies			
Mission	Adaptability	Involvement	Consistency
Creates shared vision Defines strategies and direction Defines goals	Creates changes Promotes organisational learning Customer/patient focus	Empowerment Team organisation Organisational capability	Core values Works to reach engagement Coordination
Core remote competencies Credibility Communicativeness Self-development Digital readiness			

Source: Own elaboration based on The Denison Leadership Development 360 Survey; https://www.denisonconsulting.com/wp-content/uploads/2019/06/the-denison-leadership-development-survey.pdf (accessed on 12 September 2020); [62].

**Table 2 ijerph-19-14388-t002:** The results of the normality of distribution and descriptive statistics.

Variables	KS	SW	M	SD	s.e.	MIN	MAX
Leadership strategies INVOLVEMENT	0.14	0.89	4.44	1.26	0.02	0.00	6.00
Leadership strategies CONSISTENCY	0.14	0.89	4.46	1.25	0.02	0.00	6.00
Leadership strategies ADAPTABILITY	0.14	0.89	4.47	1.20	0.02	0.00	6.00
Leadership strategies MISSION	0.14	0.89	4.44	1.23	0.02	0.00	6.00
Core remote competencies	0.12	0.89	4.54	1.15	0.02	0.00	6.00
Affective well-being	0.03	0.99	4.31	0.77	0.01	1.33	6.11

For KS, *p* < 0.01; for SW, *p* < 0.001.

**Table 3 ijerph-19-14388-t003:** Spearman’s correlations between all leadership strategies, core remote competencies and affective well-being.

Trait of Leadership	Variables	Affective Well-Being
Involvement	Empowerment	0.41 ***
	Team orientation	0.37 ***
	Organisational capability	0.36 ***
Consistency	Inspires core values	0.33 ***
	Works to reach engagement	0.29 ***
	Coordination	0.42 ***
Adaptability	Creates changes	0.40 ***
	Promotes organisational learning	0.37 ***
	Customer/patient focus	0.42 ***
Mission	Creates shared vision	0.32 ***
	Defines strategies and direction	0.36 ***
	Defines goals	0.32 ***
Core remote competencies	Communicativeness	0.38 ***
	Credibility	0.36 ***
	Self-development	0.22 *
	Digital readiness	0.35 ***

* *p* < 0.05, *** *p* < 0.001.

**Table 4 ijerph-19-14388-t004:** Spearman correlations between leadership effectiveness traits, core remote competencies and affective well-being.

Variable	Involvement	Consistency	Adaptability	Mission	Core Remote Competencies
Affective well-being	0.5 ***	0.49 ***	0.49 ***	0.48 ***	0.51 ***

*** *p* < 0.001.

**Table 5 ijerph-19-14388-t005:** Determination of the normality of variable distribution in the model and the analysis of predictor collinearity.

Variable	KS	SW	VIF
Affective well-being	0.06 *	0.99 *	NA
Leadership strategies INVOLVEMENT	0.13 *	0.89 ***	3.42
Leadership strategies CONSISTENCY	0.15 *	0.89 ***	5.72
Leadership strategies ADAPTABILITY	0.14 *	0.88 ***	4.53
Leadership strategies MISSION	0.15 *	0.89 ***	3.73
Core remote competencies	0.16 *	0.90 ***	5.55

* *p* < 0.05, *** *p* < 0.001.

**Table 6 ijerph-19-14388-t006:** The influence of independent variables on affective well-being.

Variable	B	s.e.	t	β	*p*
Constant	3.57	0.21	16.62	0.00	<0.001
Involvement	0.11	0.08	1.41	0.22	>0.05
Consistency	−0.06	0.10	−0.57	−0.11	>0.05
Adaptability	0.23	0.09	2.39	0.42	<0.05
Mission	−0.03	0.09	−0.32	−0.05	>0.05
Core remote competencies	−0.08	0.11	−0.74	−0.14	>0.05

## Data Availability

The data are available at 10.6084/m9.figshare.21200336.

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
