# Peer review of "Facing the Post-Pandemic Challenges: The Role of Leadership Effectiveness in Shaping the Affective Well-Being of Healthcare Providers Working in a Hybrid Work Mode"

_ijerph, 2022, doi:10.3390/ijerph192114388_

Round 1
Reviewer 1 Report
ABSTRACT
I found your abstract simple but explanatory.
INTRODUCTION
Relevance of the paper emerges in this phase, but it is not properly structured as a valuable scientific paper thus readability looks to be scarce.
I would recommend some important structural suggestions.
1) Despite you state that research has not focused on your issue yet, you should report authoritative citations that highlight need for research like yours (hopefully in their space for future research).
2) To help the readers, even graphically, to better focus on your proposition, you should clarify your proposal to fill the gap by means of specifically stated research questions (RQ1, RQ2, etc.) or pointed objectives. Likely you could insert it at line 86-87, after having explained research gaps you aim to fill.
3) After stating RQs, valuable scientific articles report hints about i) the methodology they will use to answer such RQs, ii) main findings, iii)managerial and academic implications of the whole paper. Please add these steps thus your paper will assume a much higher relevance and readability.
4) Please add in the introduction (at the end) a quick overview of sections you are going to discuss in the rest of the paper.
As introduction represent a very crucial phase of valuable scientific works, to address these issues, while understanding the importance of this part, you may generally find inspirations in:
- White, P. (2017). Developing research questions. Bloomsbury Publishing.
THEORETICAL FRAMEWORK AND METHODOLOGY
Theoretical part is clear and well rooted into theoretical concept. Methodology is well used but poorly explained thus I have an important suggestion.
As you cite you used several methodological approaches (statistical, literature review, survey, ecc.), I strongly suggest clarifying what these methodologies pursue and how these methodologies fit all together. The best option in my opinion is to provide a clear figure. I suggest to carefully consider the Figure 1 of:
- Hristov, I., Camilli, R., & Mechelli, A. (2022). Cognitive biases in implementing a performance management system: behavioral strategy for supporting managers’ decision-making processes. Management Research Review.
CONCLUSIONS, DISCUSSION, LIMITATIONS
This part is quite exhaustive, anyway I have a major comment.
I find implications (both academic and managerial) almost inexistent. Your work may potentially express important implications for both academics and managers. Please add a relevant paragraph on it. Here, below you can find some reference useful to develop your study.
- Hristov, I., Cimini, R., & Cristofaro, M. (2022). Assessing stakeholders’ perception influence on companies’ profitability: evidence from Italian companies. Production Planning & Control, 1-15.
- Lovallo, D., Brown, A. L., Teece, D. J., & Bardolet, D. (2020). Resource re‐allocation capabilities in internal capital markets: The value of overcoming inertia. Strategic Management Journal, 41(8), 1365-1380.
GENERAL COMMENTS
The paper is written in simple but understandable language.
I hope my comments and suggestions will be helpful in the further development of this study.
Author Response
Response to the Reviewer 1
Dear Reviewer, we highly appreciate your thoughtful comments, which do indeed helped us improve our article. Here are our responses:
|
Reviewer’s comment |
Authors’ reply |
|
1) Despite you state that research has not focused on your issue yet, you should report authoritative citations that highlight need for research like yours (hopefully in their space for future research). |
We have added a paragraph on a need to expand research into remote leadership in healthcare, based on citations (lines 86-95) |
|
2) To help the readers, even graphically, to better focus on your proposition, you should clarify your proposal to fill the gap by means of specifically stated research questions (RQ1, RQ2, etc.) or pointed objectives. Likely you could insert it at line 86-87, after having explained research gaps you aim to fill. |
We have added two research questions, after explaining the research gap (lines 102-104), as it was suggested by the Reviewer. |
|
3) After stating RQs, valuable scientific articles report hints about i) the methodology they will use to answer such RQs, ii) main findings, iii)managerial and academic implications of the whole paper. Please add these steps thus your paper will assume a much higher relevance and readability.
|
All of those steps are added after stating the research questions (lines 104-140). |
|
4) Please add in the introduction (at the end) a quick overview of sections you are going to discuss in the rest of the paper.
|
We added that overview at the end of the introduction, as it was suggested by the Reviewer. |
|
As you cite you used several methodological approaches (statistical, literature review, survey, ecc.), I strongly suggest clarifying what these methodologies pursue and how these methodologies fit all together. The best option in my opinion is to provide a clear figure.
|
We added a figure to illustrate the research design of our study along with a short section Research Design (after stating hypotheses) |
|
I find implications (both academic and managerial) almost inexistent. Your work may potentially express important implications for both academics and managers. Please add a relevant paragraph on it. |
We have added implications at the end of the discussion section. |
Reviewer 2 Report
The manuscript deals with the dimensions of leadership towards health workers' emotional well-being during remote work mode.
The research design is robust, the hypotheses are well formulated, and the regression models are clear. Hey the organization of the introduction and discussions are consistent and offer insights into the exploration of an understudied area such as leadership in digital health.
I suggest acceptance of the manuscript after minor revisions:
1. The authors state that they interviewed healthcare workers from 7 countries but it is unclear whether they used an English-language protocol, thus targeting workers fluent in English, or whether you provided multiple language versions of the questionnaires (page 7; line 302).
2. When authors describe statistical analyses, punctuation is completely missing (Page 9; lines 366-371).
3. Line 323 Leadership effectiveness strategies-please specify what low (0) or high (6) scores indicate and how to interpret low and hight total results.
4. Line 342 Affective well-being - please indicate how many items you administred in total. Do all items have the same likert scale? or you had to transform scales, scores or reverse items? Indicate minimum and maximum total, and what low and high scores indicate.
5. In Table 5 (line 433) it would be better to make the columns more homogeneous: shrink the three columns related to numbers, and make the text strings on one line/row.
6. When the authors describe the names of the statistical tests performed the font size should be revised, as it seems much larger than the rest of the text (lines 369 - 380 - 381 - 423 - 427 - 428).
7. The wording "conflict of interest" is not in bold, as it should be (page 15 line 620).
The research design and very well thought out, structured and coherent, the statistical analyses are appropriate, although future studies could confirm the proposed models through a structural analysis of the SEM model. The reliability data with Cronbach's alpha are convincing.
Author Response
Response to the Reviewer 2
Dear Reviewer, we highly appreciate your positive feedback on our article and all comments. Here are our responses:
|
Reviewer’s comment |
Authors’ reply |
|
1. The authors state that they interviewed healthcare workers from 7 countries but it is unclear whether they used an English-language protocol, thus targeting workers fluent in English, or whether you provided multiple language versions of the questionnaires (page 7; line 302). |
We have described what language versions were used in the procedure section (page 7, lines 314-315.) |
|
2. When authors describe statistical analyses, punctuation is completely missing (Page 9; lines 366-371). |
We corrected it, thank you. |
|
3. Line 342 Affective well-being - please indicate how many items you administred in total. Do all items have the same likert scale? or you had to transform scales, scores or reverse items? Indicate minimum and maximum total, and what low and high scores indicate.
|
We added all relevant information to the Measures section, under subsection “affective wellbeing” |
|
4. In Table 5 (line 433) it would be better to make the columns more homogeneous: shrink the three columns related to numbers, and make the text strings on one line/row.
|
We corrected it, thank you. |
|
5. When the authors describe the names of the statistical tests performed the font size should be revised, as it seems much larger than the rest of the text (lines 369 - 380 - 381 - 423 - 427 - 428).
|
We corrected it, thank you. |
|
6. The wording "conflict of interest" is not in bold, as it should be (page 15 line 620).
|
We corrected it, thank you. |
|
7. future studies could confirm the proposed models through a structural analysis of the SEM model |
We also added that to limitations and further research section, thank you. |